# The Nature of Global Green Finance Standards—Evolution, Differences, and Three Models

Christoph Nedopil [1,*], Truzaar Dordi [2] and Olaf Weber [2]

[1] Central University of Finance and Economics, IIGF Green BRI Center, 62 South College Road, Beijing 100081, China
[2] Waterloo University, School of Environment Enterprise and Development, Environment 3, Waterloo, ON N2L 3G1, Canada; tdordi@uwaterloo.ca (T.D.); oweber@uwaterloo.ca (O.W.)
\* Correspondence: nedopil@alumni.harvard.edu

**Abstract:** (1) Background: Green finance standards have proliferated with much need for harmonization to accelerate global green financial flows. However, little is known on the nature of green finance standards that accelerates differentiation, rather than harmonization. Therefore, we embark to answer the question what the nature of green finance standards is and specifically how green finance standards have evolved in major economic systems driven by different actors and leading to differences and commonalities over time and environmental focus area. (2) Methods: To analyze the question, we build a model based on institutional and standards theory and apply text analysis and statistical methods to analyze 84 green finance standards issued from 1998 to 2020. (3) Results: we find clear evidence that green finance standards evolve depending on economic governance types (e.g., market-based, government-based and in weak institutional environments), environmental focus areas (e.g., pollution, climate, biodiversity) and depend on actors in government, intermediaries and developing financial institutions. We also show that this development has been dynamic over the last few decades. We further test and confirm three models of green finance standards: output-based, input-based and process standards that have evolved. With the findings, we aim to provide a better foundation for both research and policy in future green finance standard research, development and harmonization.

**Keywords:** green finance; standards; institutional theory; China; European Union; emerging economies

## 1. Introduction

Green finance codes, taxonomies, regulations, guidelines, safeguards and catalogues for green finance (henceforth referred to as "standards") aim to support governments, investors, corporations and other involved stakeholders to channel finance and investments into ecologically friendly economic development, e.g., in renewable energy, sustainable transport or sustainable agriculture.

As an important foundation for the financial sector to support achieving the sustainable development goals (SDGs) and the Paris Agreement [1,2] through "the financialization of green development" [3,4], green finance standards have proliferated over the past few decades [2,5,6]. This has led to competition of green finance standards, with claims of some countries or institutions to be most advanced in green finance standard setting, hoping for others to converge to their standards [7–9].

However, "since the market for standards is unlikely to support a great variety of competing and overlapping initiatives in the long run" [5], research and policy is concerned with how to "harmonize" (rather than simply converge) various green finance standards that would allow more seamless flow of capital into different jurisdictions with lower transaction costs [10].

Such a harmonization effort, however, risks failing without a sound understanding of the reasons why heterogeneity in green finance standards exists in the first place—as path

dependency would be an impediment to standard harmonization [11]. Selected researchers have looked at drivers of adaptation of green finance standards [12], or the effectiveness for environmental protection of voluntary financial standards [13] to find that voluntary standards would "not be able to transform the global economy at large" [14]. Yet, the question of why different types and what different types of green finance standards exist or in other words "what is the nature of global green finance standards in regard to their evolution and application" is unanswered. Without an understanding of underlying differences and development paths of existing green finance standards, future harmonization will fail as the fundamental building blocks and motivations of different green finance standards remain misunderstood [15], both in theory and practice.

This study therefore aims to answer this question: what is the nature of green finance standards, and specifically what are international specificities of green finance standards with regard to their development in different governance systems and across time with regard to their content, models and their application?

We base our analysis on institutional theory [16–18] and standards theory [19], and build a model that allows us to analyze how green finance standards evolve over time and explain their differences in three different types of economic governance systems [20] relevant for green finance application: countries with market-economic governance systems, countries with government-based economic governance systems, countries with "weak" institutional economic governance [21]. By applying statistical methods and text analysis of 84 standards issued between 1998 and 2019 in the EU as a market-based economy, in China as a government-based economic governance, and in emerging economies, we find statistically significant differences of green finance standard development paths, drivers and their application in market-based, government-based and weak institutional economic governance systems. For example, we find that standards developed by developing finance institutions (DFIs) have focused more on biodiversity finance, while standards issued by governments and intermediaries have focused on climate and pollution issues. We also develop and test three models of green finance standards that are applied in different governance systems: output-based, input-based and process standards. Output-based standards aim to ensure that de-facto outcomes of investments are ecologically friendly, e.g., through provision of emission thresholds and applied particularly in the EU; input-based standards, such as catalogues, that facilitate investor decisions for "green" projects without requiring measurement of de-facto environmental impact (particularly in China); and finally process standards that provide investors with processes that improve environmental compliance, e.g., through safeguards, particularly applied in emerging economies.

As this study confirms how local specificities and pressures lead to different green finance standards and requirements, we cannot and should not try to answer the question, which green finance standard is the "best" in terms of mobilizing green finance for de-facto green development rather than "greenwashing" [22] or "green labeling" of existing investments without additionality [4].

With a better understanding on the nature of green finance, we hope to help green finance policy, green finance practice and research on several levels: On the policy level, the study helps to better understand reasons for heterogeneity in sustainable finance standards that need to be considered for further harmonization efforts of green finance standards. Based on the findings, we suggest taking local specificities of governance and environmental factors into account for potential future green finance standard harmonization.

By approaching the standard analysis as a product of "institutional work", similar to Slager et al. [18], this research also expands the academic theory applied for green finance studies: both institutional theory and standards theory have been used sparingly in green finance studies, despite their obvious relevance. Our work contributes to the green finance and sustainable finance theory, particularly by developing and proving the three green finance models (input, output, process). Those can be applied and studied further, e.g., with regard to their efficacy for an effective "financialization" of green development (e.g., [3,4]).

The remainder of the paper is structured as follows. We start by analyzing the relevant literature of institutional theory and green standard development to elaborate on our research objective and hypotheses. The next section describes the materials and statistical methods. The fourth part presents the results and the paper finishes with a discussion.

## 2. Literature Review

The green finance "standards market" [8] has been fueled by a multitude of stakeholders issuing green finance codes: governments and regulators have issued over 390 green finance measures according to the World Bank's Green Finance Platform [23]; private and public financial institutions have issued local and international standards related to green finance (e.g., International Finance Corporation (IFC) Performance Standards, Inter-American Development Bank (IADB) Environmental and Safeguards Compliance Policy, Barclays Impact Eligibility Framework for Shared Growth Ambition or HSBC's Sustainability Risk Policy); moreover non-financial and non-government institutions continue to issue standards for various aspects of green finance, including UN organizations (e.g., UN Principles of Responsible Investment (PRI)), think tanks (e.g., CICERO, Climate Bonds Initiative (CBI)), associations (e.g., Global Reporting Initiative (GRI), the Sustainable Accounting and Standards Board (SASB), the Taskforce for Climate-Related Financial Disclosure (TCFD), International Capital Market Association (ICMA), Global Impact Investing Network (GIIN)), and NGOs (e.g., Impact Management Project—IMP, Carbon Disclosure Project—CDP), as well as many labels to signal greenness of finance and investments for costumers and investors [4]. While these standards address ever more financial instruments for debt and equity, for projects and funds, as well all different investment phases, like fund raising, risk management, reporting, they all aim to accelerate more ecological-aligned—that is "green"—finance [5,24]. This should accelerate the "financialization of green development"—that is for the financial sector and its institutions to take an increasing role and responsibility shaping green development [3,4].

To analyze this multitude of green finance standards, we pool them based on their common purpose of accelerating financial flows into investments understood as ecologically desirable (e.g., to reduce pollution, create an emission-free economy, and/or improve biodiversity) [25], similar to Gilbert et al. [5] in their study on international accountability standards.

To analyze green finance standards and their nature, we first draw on the literature on the nature of standards in general. Standards facilitate coordination by defining the appropriate attributes of the standardized subject, enable external inspection and sanctioning of non-compliance [18], and are backed up by external bodies, such as professional associations or governments [19]. As such, green finance standards provide common understanding, frameworks, languages or procedures to a defined set of stakeholders to enable economic exchange with lower transaction costs than without such standards, e.g., by defining what is "green" (e.g., Chinese green bond catalogue, EU Taxonomy), how to evaluate the "green risks" of a project over many projects (e.g., Equator Principles), or how to report on the impacts of green investments (e.g., Global Reporting Initiative, GRI) [26].

To understand the "causes and consequences" of standard development, examine the evolution and development of standards [18] and "to examine the diffusion of standards" [19], institutional theory has offered important foundations. Institutional theory examines how institutions are created, maintained or disrupted "by the purposive and practical actions of individuals and organizations" [17,18]. Within this theory, institutions are defined as rules, regulations, standards [27] that in an economy determine the costs of exchange [28]. Accordingly, green finance standard development can be analyzed through the lens of institutional theory due to them being influence by and their influence on the exchanges between regulators, capital markets including financial institutions, investees including project owners, auditors, and a wider stakeholder group including the public (e.g., [29–33]). Institutional theories thus are well suited to study the evolution, differences, and application of green finance standards.

By applying institutional theory to analyze the nature of green finance standards, we expand previous studies applying institutional theory in this field: Solomon [34] provides an explanation of differences in carbon pricing using institutional theory, while Branch [35] applies it to analyze aspects of behavioral finance; Guild [36] uses it as a "flexible framework for probing impediments to green finance in Indonesia in a structured and empirically rich way", while Gilbert et al. [5] apply it to understand the emergence of accountability standards; Slager et al. [18] analyze emergence of responsible investment standards using institutional theory, while Mosyński and Wieckowska [37] build a theoretical model study of the evolution of ecological financial markets from the perspective of institutional economics.

## 3. Hypotheses

In order to analyze the evolution and difference of green finance standards, we construct four hypotheses based on institutional theory and standards theory.

### 3.1. Levels of Institutional Change

Institutional theory predicts that standards emerge on different levels—such as market, civil society, government or "invisible hand". The level of institutional change depends on the broader institutional setting (e.g., availability of public or private enforcement mechanisms) and the power arrangements between the actors [16,38,39]. Relevant actors that drive institutional development representing different levels include (1) firms, particularly not state-linked firms in market economies [40], with their aim to maximize wealth change institutions. Firms often work together with intermediary groups (particularly in societies with higher degrees of freedom) [41], such as interest groups, associations, lobbies or NGOs. These actors are encouraged to support institutional development in a "celebration of the capacity for responsible self-regulation" [4]; (2) state actors, such as regulators or executive powers will mostly change formal institutions through legislation, provision of incentives or disincentives; and (3) supra-national and development institutions, such as the World Bank, International Monetary Fund (IMF), that influence institutions according to their explicit mandate, particularly in emerging economies and finally and are seen as "the invisible hand" [16].

The influence of institutional setting driving institutional development on different levels has been confirmed, for example, by Battaglini and Harstad [42] who find that environmental agreements differ depending on the actors and power structure leading to more and less specific contracts.

**Hypothesis 1.** *The level at which green finance standards evolve depends on the institutional setting, where "evolve" refers to both development and application and institutional setting refers to the economic governance of market versus government power.*

Accordingly, we distinguish three such institutional settings: (1) countries with strong market-based mechanisms, where we hypothesize that green finance standards should emerge on the market level (e.g., firms and intermediary groups as change agents); (2) countries with strong government actors, where green finance standards will emerge on the government level with non-government actors playing a negligible role as they do not have the power to institutionalize standards; and (3) countries with neither strong market-based nor strong government mechanisms, where we hypothesize that green finance standards would have to be set—if at all—by "the invisible hand", such as non-domestic intermediaries (e.g., development finance institutions) [43]. This hypothesis is not to say that these actors do not interact and influence each other throughout the levels and across time (e.g., as through a "related perspective"; as Giamporcano et al. [44] showed, governmental actors and market actors influence each other for sustainable standards development). Rather, this hypothesis researches the main level(s) of green finance standards evolution and their respective application.

### 3.2. Focus of Institutional Change

Institutional development depends on [45–47] functional, political and social pressures [48]. These pressures are specific to their environment (e.g., the financial risks of climate change, risk of biodiversity or pollution are actualized and perceived differently in different countries) and accordingly require specific solutions.

**Hypothesis 2.** *The focus of green finance standards is evolving depending on the a) actor and b) over time in regard to the three main ecological aspects of green finance as defined, e.g., by the Green Development Guidance for BRI projects: pollution, biodiversity, and climate [49].*

### 3.3. Proliferation of Institutional Change

Institutional theory explains that once a problem becomes more broadly recognized and defined, institutional development will accelerate and proliferate as more actors will become involved [8,50]. Chiapello [4] argues that green finance (and in extension their standards) is the "fourth moment in environmental policies", following the invention of environmental policies in the 1960s, the promotion of market instruments in the 1980s and the increasing role of corporate social responsibility around the turn of the century. Therefore, with a broader recognition of the role of green finance, green finance standard proliferation should accelerate.

**Hypothesis 3.** *Green finance standard development will accelerate the more widely recognized the need for action for green finance becomes.*

### 3.4. Models of Institutional Change

With a plethora of standards, institutional theory predicts that different models of standards should emerge, both due to competition of standards [5] and due to path-dependence, where institutional change is dependent on previously existing and different institutional settings [11], local capacity or local environmental circumstances.

Thus, rather than full isomorphism of green finance standards [51], different models for green finance standards will manifest in a "standards multiplicity" [8]. Timmermans and Berg [52] identified four models of standards: design standards that define properties and features, terminological standards that ensure stability of definitions, performance standards that set output specifications, and procedural standards that specify how processes are performed.

**Hypothesis 4.** *Different models of green finance standards will emerge (a) dependent on the actors, (b) the jurisdictions, and (c) the time specialized for their relevant environments.*

## 4. Materials and Methods

In order to test the hypotheses and study the nature of green finance standards, we collected over 200 green finance standards issued by three main actors for green finance standards between 1998 and 2019:

- Governments and regulators (e.g., green bond and green credit standards in China)
- Intermediary groups and associations (e.g., non-governmental organizations, such as ICMA, CICERO, Climate Bonds Initiative and multilateral organizations, such as UN organizations dealing with green finance)
- Supranational and development institutions including multilateral and bilateral development financial institutions (e.g., IFC, Asian Development Bank (ADB), Asian Infrastructure and Investment Bank (AIIB))

We chose markets with more bank-based financial systems compared to market based systems—that is systems where banks play a leading role in mobilizing savings, allocating capital, overseeing the investment decisions, etc., as compared to securities

markets [53,54]. Furthermore, we chose three different types of green finance markets with different governance forms, according to the Index of Economic Freedom [55]:

(1).　Market economies—where we chose the European Union (EU): Much of the development of the EU's green finance system was driven by market stakeholders (particularly banks, and to some extent securities markets and shareholders): the German development bank KfW claims, for example, to have "supported green finance" through its environmental program for SME's already in 1984 [56]. Besides the market-driven development, the EU and its nations, had been developing green finance standards on the government level to regulate green finance with the first related green finance standard published in 1998, and 113 more since then [23]. The EU's environmental strategy is based on, e.g., the Gothenburg Summit in 2001, where it called for a "new approach to policy making that ensures the EU's economic, social and environmental policies mutually reinforce each other" with a special mention of climate change. In 2007, The European Council "insisted on the need to give priority to implementation measures" that included the "protection of biodiversity and ecosystem services" and "calls upon business, NGOs and citizens to become more involved in working for sustainable development" [57]. Yet, few comprehensive government-led green finance standards had so far been issued, which brought even advanced EU economies, like Germany, to the conclusion that the regulatory approach for green finance has trailed [58]. To accelerate its regulatory approach, the EU introduced the Sustainable Finance Action Plan in 2018 [59] and published the EU Taxonomy on Sustainable Activities" (the de-facto EU green and sustainable finance standard) in 2019 [60] (which, in March 2021, was still waiting for the final approval). Similarly, the European Central Bank had only seriously considered climate and environmental risks in its "Guide on climate-related and environmental risks for banks" published in November 2020 [61]. Despite little government regulation, the EU has become one of the largest markets for green finance, for example with more than USD 125 billion in green bonds issued in 2020 [62].

(2).　Government-led economies, where we chose China: As a country with strong government influence in the economy and with weaker scores with regard to free market mechanisms (e.g., [55]), China's green finance development has seen multiple government-level green finance interventions [63], particularly by Cinese Banking and Insurance Regulatory Commission (CBIRC)—the Chinese banking regulator (e.g., Green Credit Guidelines in 2012, Green Credit Statistics System in 2013 and Green Credit Key Performance Indicators in 2014) for the banking sector, and by the People's Bank of China (e.g., Guidelines for Establishing the Green Financial System in 2016, the Green Bond Catalogue in 2016) [63] to regulate the green bond market. By 2019, 12% of the recorded sustainable finance standards issued by governments recorded on the World Bank's Green Finance Platform were issued by China—making China the most active green finance regulatory standard issuer in the world. Meanwhile, market-driven initiatives had been few (an exception, was the introduction of the Green Investment Principles for the Belt and Road Initiative in 2018 [64]). China's green finance system is embedded in and supporting national strategies that focus on the "three key battles" of poverty alleviation, air pollution and financial stability [65], and supports the "battle for blue skies" [66]. Its green finance system therefore has a strong focus on fighting air pollution (which led to the inclusion of "clean coal" in the Chinese Green Bond Catalogue and Green Industry Catalogue, which has little air pollution, but high greenhouse gas emissions). To accelerate climate finance, relevant ministries issued a separate climate finance guidance [67] in November 2020. As a consequence of much government support [68], China has become one of the largest markets for green finance [69] with USD 800 billion in green bonds issued over the past years [70], and about USD 1.85 trillion of outstanding green credits [71].

(3).　Emerging economies with neither particularly strong market nor government-led governance: with less developed domestic financial market and with a higher dependence

on foreign aid in many of the emerging economies, an important driver for green finance standards evolution and application has been developing finance institutions (DFIs), such as multilateral development banks (e.g., the IFC, World Bank, Asian Development Bank, European Bank for Reconstruction and Development (EBRD)) or bilateral development banks or financial programs (e.g., the French Development Bank AFD, the British Department for International Development (DFID) or the German Development Bank KfW). The development and application of green finance standards in weak governance countries became necessary, as DFIs were expected by their investor countries (mostly developed countries) to contribute to sustainable development and sustainable investments. To overcome a lack of market-driven and government-driven green finance standards in less developed countries, DFIs developed their own green finance standards, such as IFC's Environmental and Social Review Procedure (ESRP) from 1998. This procedure was updated in 2006 with IFC's Sustainability Framework. IFC's Performance Standard Framework was published in 2012 and has since been widely adopted and adapted by other DFIs (e.g., ADB, KfW, European Investment Bank (EIB)) investing in emerging economies. With a particular financial sector development mandate, IFC, supported green finance standards integration in many of the emerging market financial institutions they invested (e.g., through adopting the Equator Principles) and supported emerging market regulators to develop and apply green finance standards through the Sustainable Banking Network (SBN) it helped establish in 2012 [72].

For the collection of green finance standards in these three economic governance systems, we used the World Bank's Green Finance Platform, and collected many standards by hand. For our analysis, we collected each iteration of the selected green finance standards (such as five iterations of the Global Reporting Initiative) to trace historical development of standards, similar to Reinecke et al. [8] in their study on sustainability standards in the coffee industry.

We labelled each standard regarding the type of the issuer (e.g., government, development financial institution, intermediary), the year they were published, and the dominant application jurisdiction distinguished between "EU", "China" and "emerging economies" (e.g., the application jurisdiction for the Equator Principles was labelled as "EU" and "emerging economies", but not "China", as by 2017 only one Chinese bank had signed up to the Equator Principles).

Particularly for standards issued by non-government actors, we included green finance standards issued with a purpose of broader application potential and thus the goal to lead to institutionalization beyond a single financial institution. We therefore did not include green finance standards issued by individual financial institutions which tend to be for internal use (e.g., a bank's individual ESG framework).

After checking for completeness and jurisdiction, as well as application relevance (based on a number of interviews with financial sector specialists), the final list of full texts that we analyzed includes 84 green finance standards (see Table 1): 16 from development financial institutions, 28 from multilateral institutions (where some institutions, such as IFC, are labelled as both financial institutions and multilateral institutions, the difference is tested for in the analyses), 52 from NGOs and 9 government/regulatory standards issued between 1998 and 2020.

**Table 1.** List of sustainable finance standards from different stakeholders.

| Name of Issuer | Financial Initiative | Year |
|---|---|---|
| International Finance Corporation (IFC) | Environmental and Social Review Procedure (ESRP) (predecessor of IFC Sustainability Framework in 2006) | 1998 |
| Global Reporting Initiative (GRI) | Global Reporting Initiative Guidelines G1 | 2000 |
| GRI | Global Reporting Initiative Guidelines G2 | 2002 |
| Carbon Disclosure Project (CDP) | Carbon Disclosure Project (CDP) | 2002 |
| Asian Development Bank (ADB) | Environment Policy | 2002 |
| Equator Principles | Equator Principles EP I | 2003 |
| Extractive Industries Transparency Initiative (EITI) | Extractive Industries Transparency Initiative Standard | 2003 |

**Table 1.** *Cont.*

| Name of Issuer | Financial Initiative | Year |
| --- | --- | --- |
| GRI | Global Reporting Initiative Guidelines G3 | 2006 |
| Equator Principles | Equator Principles EP III | 2006 |
| United Nations (UN) | United Nations Global Compact Principles | 2006 |
| IFC | IFC Performance Standards Social and Environmental Sustainability | 2006 |
| UN | Principles of Responsible Investing (PRI) | 2006 |
| ADB | ADB Safeguard Policy Statement | 2009 |
| Climate Bonds Initiative (CBI) | Climate Bonds Initiatives—Climate Bonds Standard and Certification Scheme | 2009 |
| International Integrated Reporting Council (IIRC) | International Integrated Reporting Council (IIRC) | 2010 |
| OECD | OECD Guidelines for Multinational Enterprises | 2011 |
| International Development Finance Club (IDFC) | International Development Finance Club (IDFC) | 2011 |
| Asia Investor Group on Climate Change (AIGCC) | Asia Investor Group on Climate Change | 2011 |
| Sustainability accounting Standards Board (SASB) | Sustainability Accounting Standards Board Industry Standards | 2011 |
| IFC | IFC Performance Standards Social and Environmental Sustainability | 2012 |
| IFC | IFC Performance Standard 1—Environmental and Social Risks | 2012 |
| IFC | IFC Performance Standard 6—Biodiversity Conservation | 2012 |
| SASB | SASB Commercial Banks | 2012 |
| United Nations Environmental Programme (UNEP) | UNEP Principles for Sustainable Insurance (PSI) | 2012 |
| China Banking Regulatory Commission (CBRC) | CBRC—Green Credit Guidelines | 2012 |
| IFC | IFC—Sustainable Banking Network | 2012 |
| Global Investor Coalition | Global Investor Coalition on Climate Change | 2012 |
| GRI | Global Reporting Initiative Guidelines G4 | 2013 |
| Equator Principles | Equator Principles EP III | 2013 |
| EITI | Extractive Industries Transparency Initiative Standard | 2013 |
| ADB | ADB Safeguard Policy Statement | 2013 |
| ADB | Environment Operation Direction | 2013 |
| Green Climate Fund (GCF) | GCF Investment Framework | 2013 |
| UNEP | Partnership for Action in Green Economy (PAGE) | 2013 |
| International Capital Markets Association (ICMA) | ICMA—The Green Bond Principles | 2014 |
| CBRC | CBRC—Key Indicators of Green Credit Performance | 2014 |
| Alliance for Water Stewardship (AWS) | International Water Stewardship Standard | 2014 |
| China Social Enterprise and Investment Forum (CSEIF) | China Social Enterprise and Investment Forum | 2014 |
| Global Investor Coalition | The Low Carbon Registry | 2014 |
| Multilateral Development Banks (MDB) | MDB—IDFC Common Principles for Climate Mitigation Finance Tracking | 2015 |
| CiCERO | CICERO Shades of Green | 2015 |
| Global Infrastructure Basel (GIB) | The Standard for Sustainable and Resilient Infrastructure | 2015 |
| French Ministry for the Ecology | French Ministry for the Ecological and Solidary Transition—Greenfin Label | 2015 |
| Task Force for Climate-related Financial Disclosure (TCFD) | Financial Stability Board—Task-Force on Climate-Related Financial Disclosures | 2015 |
| Global Steering Group (GSG/GSGII) | Global Steering Group for Impact Investment | 2015 |
| GRI | Global Reporting Initiative Standards | 2016 |
| United Nationss Principles of Responsible Investments (UNPRI) | UNPRI—Private Equity Action on Climate Change | 2016 |
| People's Bank of China (PBOC) PBOC | Chinese Green Bond Catalogue | 2016 |
| TCFD | Financial Stability Board—Task-Force on Climate-Related Financial Disclosures | 2017 |
| ICMA | ICMA—Sustainability Bond Guidelines (SBG) | 2017 |
| ICMA | ICMA—The Social Bond Principles | 2017 |
| ASEAN Capital Markets Forum (ACFM) | ASEAN Green Bond Standards | 2017 |
| Network for Greening the Financial System (NGFS) | Central Banks and Supervisors Network for Greening the Financial System (NGFS) | 2017 |
| Green Investment Group | Green Investment Principles | 2017 |
| World Bank | World Bank Environmental and Social Framework | 2017 |

**Table 1.** *Cont.*

| Name of Issuer | Financial Initiative | Year |
|---|---|---|
| SASB | SASB Investment Banking and Brokerage | 2018 |
| SASB | SASB Commercial Bank Standard | 2018 |
| ICMA | ICMA—The Green Bond Principles | 2018 |
| CICERO | CICERO Shades of Green | 2018 |
| GIB | SURE—The Standard for Sustainable and Resilient Infrastructure | 2018 |
| GSG/GSGII | Global Steering Group for Impact Investment | 2018 |
| World Benchmarking Alliance (WBA) | World Benchmarking Alliance (WBA) | 2018 |
| ICMA | ICMA—Sustainability Bond Guidelines (SBG) | 2018 |
| World Bank | World Bank—Environmental and Social Safeguards Framework ESS 1—Environmental and Social Risk | 2018 |
| World Bank | World Bank—Environmental and Social Safeguards Framework—ESS 6 Biodiversity | 2018 |
| ASFI/WWF | Asia Sustainable Finance Initiative | 2018 |
| Loan Market Association (LMA) | LMA—Green Loan Principles | 2018 |
| Science-Based Target Initiative (SBTI() | Science-Based Targets Initiative for Financial Institutions (SBTI FI) | 2018 |
| Carbon Disclosure Project (CDP) | Carbon Disclosure Project (CDP) | 2019 |
| EITI | Extractive Industries Transparency Initiative Standard | 2019 |
| IFC | IFC Performance Standard 6—Biodiversity Conservation | 2019 |
| UN | Principles of Responsible Investing (PRI) | 2019 |
| CBI | Climate Bonds Initiatives—Climate Bonds Standard and Certification Scheme | 2019 |
| AWS | International Water Stewardship Standard | 2019 |
| French Ministry for the Ecology | French Ministry for the Ecological and Solidary Transition—Greenfin Label | 2019 |
| UNDP | SDG Impact Practice Standards for Private Equity Funds | 2019 |
| Global Impact Investment Network (GIIN) | IRIS+ | 2019 |
| LMA | LMA—Sustainability Linked Loan Principles | 2019 |
| UNPRI | Inevitable Policy Response (IPR) | 2019 |
| ICMA | ICMA Harmonized Framework for Impact Reporting | 2019 |
| European Union (EU) | EU—Sustainable Finance Taxonomy | 2019 |
| 2∞investing initiative | Paris Agreement Capital Transition Assessment (PACTA) | 2019 |
| GIP | Green Investment Principles for the BRI | 2019 |
| Equator Principles | Equator Principles EP IV | 2020 |

To test the hypothesis with the data, we use qualitative and quantitative analysis. For the quantitative analysis, we adopted advancements in quantitative text mining and statistical analysis of unstructured text data. This research design advances previous studies of standards that used qualitative approaches exclusively, such as interviews, observations and mechanisms-based theorizing based on authors' subjective opinions (e.g., [8]) or "constant comparison" (e.g., [18]). Though still nascent in management literature, text mining methodologies have been used to examine trends in sustainability reports and corporate disclosure [73]. Our analysis is conducted using the open-source R software and several packages, including Tidytext [74], and tm [75]. Statistically, we test for non-random association between categorical variables, by applying the chi-square goodness of fit test (Equation (1)). In instances where the sample size is too small for approximation, the Fisher exact test will be applied instead (Equation (2)). The effect size is measured using Cramer's V (Equation (3)). Finally, Pearson residuals (Equation (4)) are applied to identify the deviation of the expected and observed values for each cell.

Equation (1): Chi-Square Test

$$\chi_c^2 = \sum \frac{(O_i - E_i)^2}{E_i} \tag{1}$$

where $O_i$ = Observed Frequencies, $E_i$ = Expected Frequencies

Equation (2): Fisher's Exact Test

$$p = \frac{\left(\frac{n_{1,1}+n_{1,2}}{n_{1,1}}\right)\left(\frac{n_{2,1}+n_{2,2}}{n_{2,1}}\right)}{\left(\frac{n_{1,1}+n_{1,2}+n_{2,1}+n_{2,2}}{n_{1,1}+n_{2,1}}\right)} \tag{2}$$

Equation (3): Cramer's V

$$\phi_c = \sqrt{\frac{\chi^2}{N\min(r-1; c-1)}} \tag{3}$$

Equation (4): Pearson Residual

$$X^2 = \sum \frac{\left(O_{ij} - E_{ij}\right)^2}{\sqrt{E_{ij}}} \tag{4}$$

## 5. Results

### 5.1. Hypothesis 1: The Level of Green Finance Standard Evolution Depends on the Country's Institutional Setting

To quantitatively test the hypothesis that green finance standards were published on different levels, we apply a goodness of fit test to analyze the predominant publisher of green finance standard in various economic governance settings. Turning to our sample, we examine 8 standards published by regulatory bodies, 16 reports published by development finance institutions, and 59 reports published by non-governmental and multilateral associations. By region, 35 standards relate to China, 59 relate to emerging markets, and 60 relate to the European Union. We note that some reports may cater to more than one region and thus the number may be greater than the sample size.

To test whether the level of green finance standard evolution varies by the country's economic governance system, we conduct a Fisher's exact test (Equation (1)) to identify the association between categorical variables. We find a statistically significant association between the two categories ($p = 0.006$, Cramer's V = 0.21), suggesting that there is a significant relation between the region and the publishing agent. A Pearson residuals test can inform where that association lies.

We find a statistically significant positive association (x = 2.46) between development finance institutions and emerging markets, and a statistically significant negative association (x = −2.01) between development finance institutions and the European Union. Together, these residuals contribute to 72.6 percent of the association and thus account for most of the difference between expected and observed values.

Accordingly, we confirm that the level of relevant green financial standards development is dependent on the governance system: particularly in economies with weaker governance and market mechanisms, developing finance institutions play a large role in green finance standard development, while in strong market economies (e.g., EU) intervention by developing finance institutions does not play a large role, compared to market level initiatives (e.g., PRI application). In government-led economies, the government level is the drivers for green finance standard development.

While the issuance is one part of the evolution, the other is their application by the financial sector. As we assume that laws apply for the domestic financial sector (e.g., a green finance regulation applies to its domestic jurisdiction), we need to test whether the application of voluntary standards published by intermediaries are applied throughout the different types of governance systems. Accordingly, we compare the application of two of the most relevant and widely applied global green finance standards—the Principles of Responsible Investment (PRI) and the Equator Principles: The PRI, a finance standard launched by an intermediary in 2006, promotes the integration of environmental, social and

governance factors into financial decision-making. It attracted signatories mostly from OECD and EU countries and only few from emerging economies and very few from China: Three years after its initiation in 2009, 159 of 311 PRI signatories were from EU countries (a further 52 from other OECD countries) (see Figure 1). By 2019, 1035 of the 2765 PRI signatories were from the EU (with 1399 signatories from other OECD countries), and by 2020 only 35 Chinese institutions have signed up to the PRI.

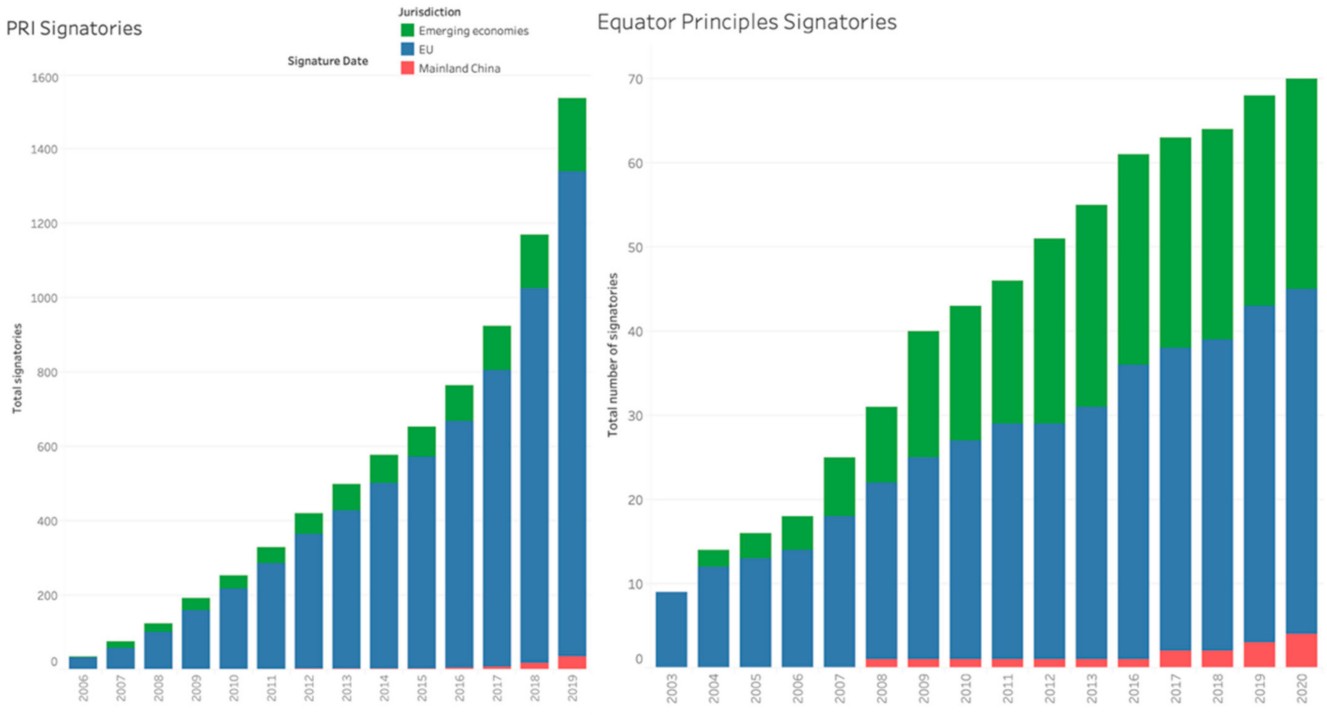

**Figure 1.** Principles of Responsible Investment (PRI) Signatories 2006–2019 and Equator Principles Signatories 2003–2019.

A similar picture emerges when looking at signatories of the Equator Principles—another market-driven green finance standard for financial institutions' project finance environmental risk management. Since its launch in 2003, 105 financial institutions from 35 countries had signed up by April 2020. The largest proportion of signatories continues to be from EU countries (41 signatories). Of the Equator Principals signatories, only 4 came from China in 2020, and 25 signatories from emerging economies.

It could be conceivable that the lower number of Chinese organizations following these voluntary standards is due to a smaller size of the financial markets and thus a lower number of financial actors. However, with China being home to over 4500 financial institutions in 2019, compared to about 5500 in the EU [76,77] and China being home to the world's four largest financial institutions by assets (Bank of China, Agricultural Bank of China, China Construction Bank and Industrial Bank of China) [78], it can be stated for China, as a government-centric governance model, voluntary green finance standard application is irrelevant.

Therefore, Hypothesis 1 can be confirmed that the level of the evolution in terms of development and application depends on the governance system with market-led countries focusing on market-level standards in their development and application, China as a government-centric system focuses on the government issued standards and does not apply market-level standards, while emerging economies have often relied on green finance standards from DFIs.

*5.2. Hypothesis 2: The Focus of Green Finance Standards Is Evolving*

The focus areas of green finance standards can broadly vary between green factors including pollution, greenhouse gas (GHG) emissions, biodiversity.

To quantitatively analyze the evolution of contents of the different standards with regard to actors, application area and time frame, we begin by identifying the frequency of a collection of words that are closely related with (1) biodiversity, based on keywords identified in the Convention on Biological Diversity (CBD) [79,80], (2) climate, based on the climate-related United Nations Framework Convention on Climate Change UNFCCC and the Intergovernmental Panel on Climate Change (IPCC) frameworks [81], and pollution, based on the World Health Organization's (WHO) air pollution framework [82] (see Table 2).

**Table 2.** Keywords for biodiversity, climate, pollution.

| Biodiversity | Climate | Pollution |
|:---:|:---:|:---:|
| species | adaptation | air pollution |
| ecosystems | mitigation | PM2.5 |
| biosphere | temperature | particulate matter |
| bio-finance | greenhouse gas | air quality |
| extinction | GHG emissions | pollutant |
| habitat | extreme weather | lung cancer |
| conservation | drought | exhaust |
| deforestation | floods | soot |
| wetlands | storms | sulfur oxide |
| oceans | global warming | nitrogen oxide |

5.2.1. Hypothesis 2a: Focus Area Depending on Actor

We first test for association between focus and actor (development finance institutions, associations and regulators) using Fisher's exact test. The result finds that there is a significant difference between actor and focus ($p < 0.001$, Cramer's V = 0.32). Turning to the Pearson residuals, development finance institutions place a larger focus on biodiversity and place less focus on climate. Together, those relationships contribute 66.6 percent to the total Fisher exact score. This suggests that DFI's are putting greater emphasis on biodiversity than would be expected. To test for robustness, we removed the IFC Performance Standard 6, which focuses on biodiversity, and reconducted the Fisher exact test. The results remain significant without the IFC Performance Standard 6 ($p < 0.001$). Moreover, regulators and associations have a significant negative association with biodiversity and a significant positive association with climate (see Figure 2).

5.2.2. Hypothesis 2b: Focus Area Depending on Jurisdiction

We also test for association between focus and jurisdiction using Fisher's exact test. The result suggests that there is a significant difference between region and focus ($p < 0.001$, Cramer's V = 0.22). We find a statistically significant positive association between emerging markets and biodiversity and a statistically significant negative association between emerging markets and climate. Together, the residuals between emerging markets and focus account for 51.8 percent of the total Fisher exact score. We also find that discourse on biodiversity is less associated with China and the EU, whereas discourse on climate is more associated with both. Pollution is not significantly associated with any specific region (see Figure 3).

This comes somewhat as a surprise, as we find, for example, that several of China's green finance standards (2016 Green Bond Catalogue, 2019 Green Industry Catalogue) encourage investments in "clean coal" fired power plants, with the goal to reduce air pollution, while it does not count as a climate-friendly investment [83].

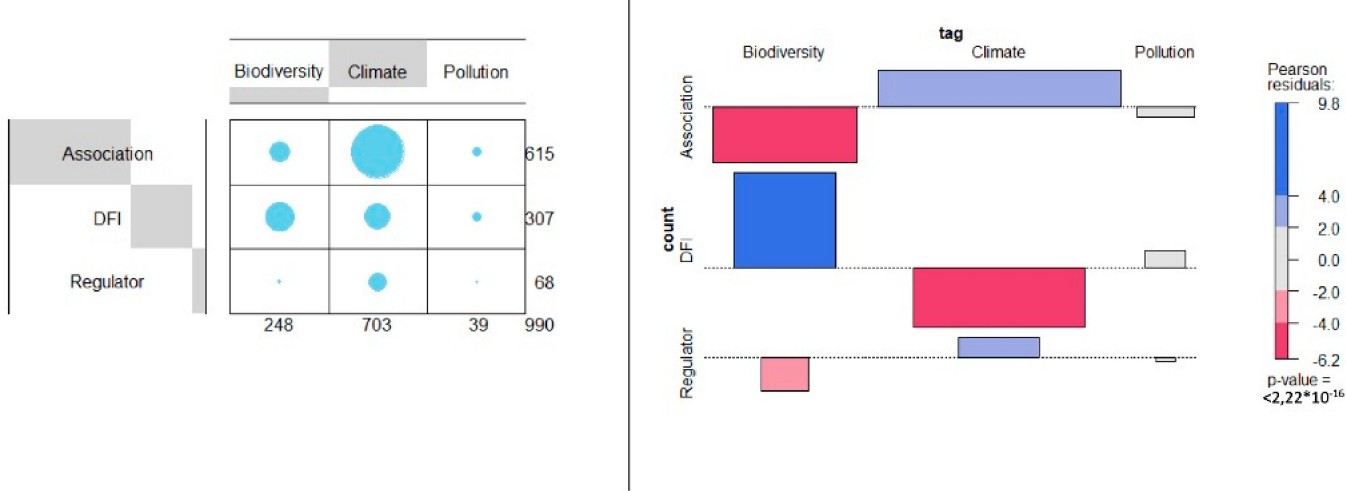

**Figure 2.** Fisher's Exact Test for Hypothesis 2a—focus area and actor. It shows the frequency of biodiversity-, climate- and pollution-related terms issued by different issuers and calculates the Pearson's residual, showing for example that "biodiversity"-related green finance standards are significantly negatively correlated with "associations", and significantly positively correlated with developing finance institutions (DFIs).

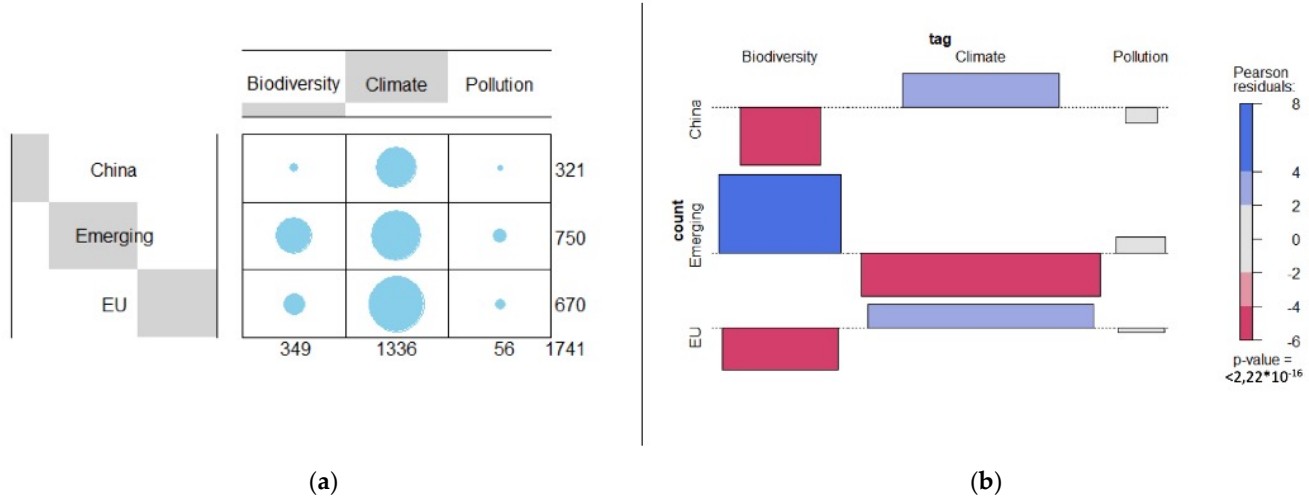

| (**a**) | (**b**) |

**Figure 3.** Fisher's Exact Test for Hypothesis 2b—focus area and jurisdiction. It shows (**a**) the frequency of biodiversity-, climate- and pollution-related terms issued in different jurisdictions (China, emerging economies, EU) and (**b**) calculates the Pearson's residual, showing for example that "biodiversity"-related green finance standards are significantly negatively correlated with China, and significantly positively correlated with emerging economies.

### 5.2.3. Hypothesis 2c: Focus Area Depending on Time

Finally, we test for the association over time and region using the Chi-Square test. The result shows that there is no significant difference of topics between periods and focus ($p = 0.123$, Cramer's V = 0.06) (see Figure 4).

Overall, Hypotheses 2a and 2b can be confirmed meaning that the focus of green finance standards depends on the issuer and local specificities, while we find no significant focus of relevant topics for green finance over time (Hypothesis 2c).

### 5.3. Hypothesis 3: Green Financial Standards Accelerate with Broader Recognition

In Hypothesis 3, we postulate that green finance standards will proliferate the more broadly the problem of the relationship between ecological change and finance is understood.

While the Bali Action Plan highlighted the need for "innovative ways" and for mobilizing "private and public sector funding and investments, including facilitation of

climate-friendly investments" already in 2007 [4], we find on the Green Finance Platform [23] (see Figure 5) that the proliferation green finance standards had been slow in the 1980 and 1990s and accelerated from 2010 to 2015 to about 17 measures per year. Issuances doubled in 2015 to 33 measures per year and reached a peak of 89 measures in 2019. We did a robustness test to analyze whether this acceleration of green finance-related issues is also visible in other forms and analyzed academic publications on the topics of "green finance" and "sustainable finance", indexed on the Web of Science database.

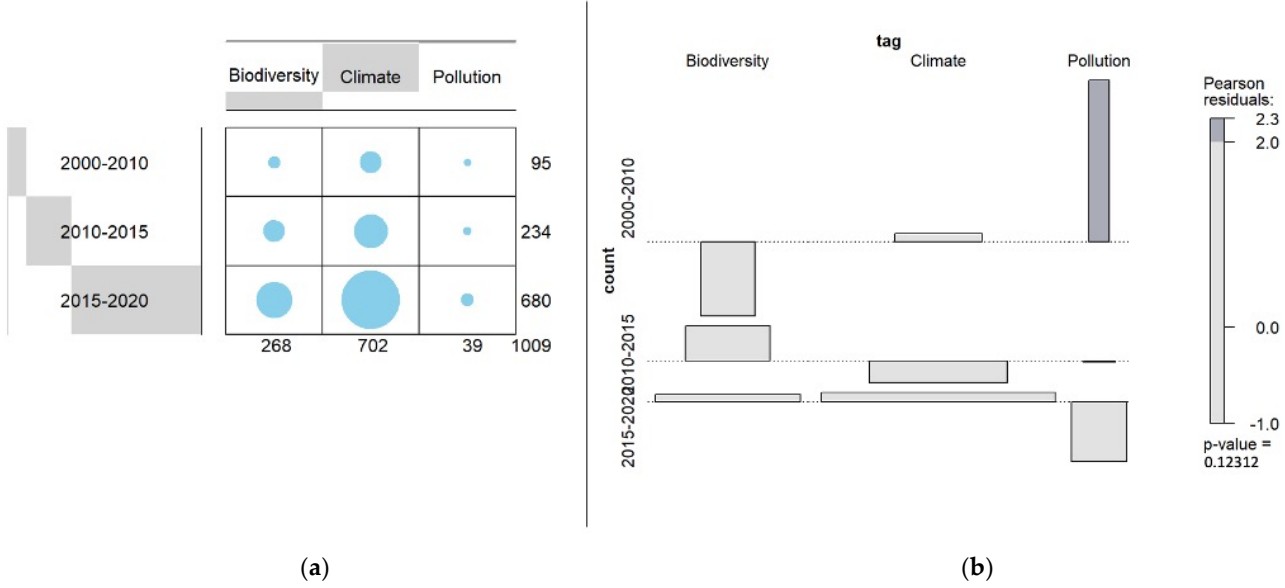

(**a**)　　　　　　　　　　　　　　　　　　　(**b**)

**Figure 4.** Fisher's Exact Test for Hypothesis 2c—focus area and time. It shows (**a**) the frequency of biodiversity-, climate- and pollution-related terms issued across time and (**b**) calculates the Pearson's residual, showing for example that only "pollution"-related green finance standards are close to being significantly correlated with early publication (visible by the lack of coloration).

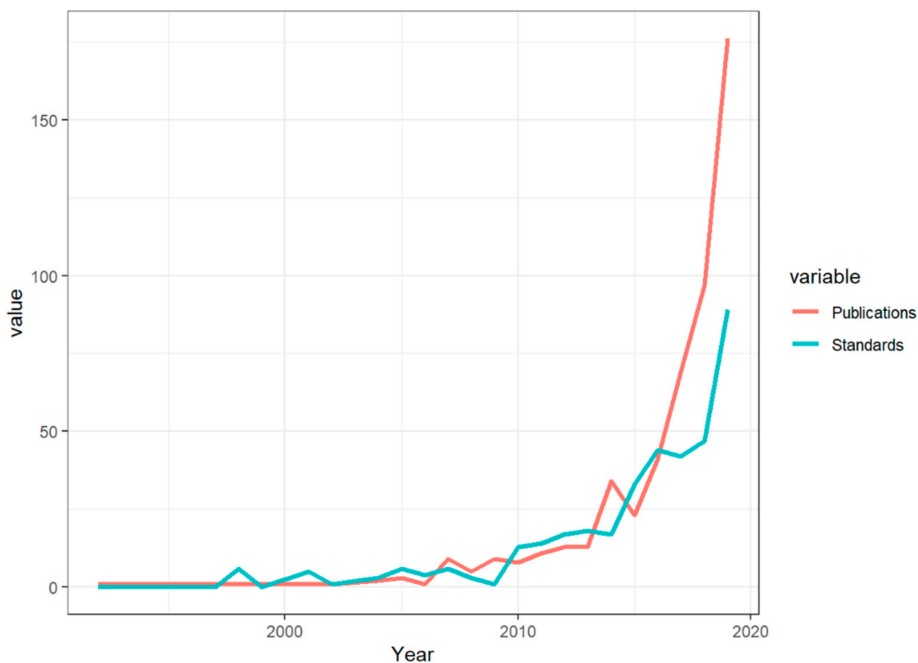

**Figure 5.** Number of Green Financial Measures (issued by Governments) and academic publications per year 1978–2019 (own depiction) (Green Finance Platform, 2020).

A Pearson's product–moment correlation test finds that developments in green and sustainable finance publications are significantly correlated with sustainable finance standards with a correlation coefficient of 0.95 ($p < 0.001$). A Welch two sample t-test finds that the average number of publications is not significantly different between peer-reviewed publications (23.6) and published standards (16.8) ($p = 0.502$).

Accordingly, we find that the year 2015 seems to be a breaking year of green finance regulation with a broader recognition of the topic. Based on induction, we find that one event can be understood as a landmark objectivation to define the responsibility of the financial sector to address green development goals: The Paris Climate Accord of 2015 signed by 195 parties. Compared to the Kyoto Protocol of 1998 (the predecessor of the Paris Climate Accord), where "finance" is mentioned one time throughout the document, the Paris Climate Accord addresses the topic of finance 15 times and puts finance already in Article 2 Section 1c of the Accord:

> "*Making finance flows consistent with a pathway towards low greenhouse gas emissions and climate resilient development*" [84]

We therefore infer that the recognition of the central role of finance in addressing climate change through the Paris agreement has led to a proliferation of green finance standards and Hypothesis 3 can be confirmed, while previous attempts to accelerate green finance have not led to a broad proliferation of green finance standards (and literature).

### 5.4. Hypothesis 4: Different Models of Green Finance Standards Will Be Developed Depending on the Problem Objectivation and Local Capacity

As green finance standards are implemented in different jurisdictions and on different levels, different models of green finance standards are expected to emerge. The standards theory [19] distinguishes between terminology standards, performance standards for output specifications, procedural standards and design standards. We know from our professional experiences in the application of different standards that safeguard procedures for green finance (e.g., IFC Performance Standards) focus on the *process* to ensure financing activities are green. A process standard can conceptually ensure sustainable alignment by guiding or forcing investors through a specific process (e.g., safeguards in the project evaluation phase, financial covenants in the project oversight phase and reporting processes), but requires only a process protocol, not green outcomes.

Another type of green finance standard provides specific project catalogues in taxonomies (e.g., Chinese Green Bond Catalogue of the PBOC, Climate Bonds Initiative standards) with clear input descriptions and prescription of specific technologies, industries and sectors that can be viewed as green and thus invested in. This standard aims to ensure that investments in specific technologies lead to green development, while it does not require a high sophistication for the verification of green "outputs" (e.g., emissions). Thus, these standards provide green "labels" for specific projects [4].

Standards that focus on the "*outputs*" of financing activities prescribe permissible environmental outputs/impacts of an investment (e.g., emissions per unit of electricity produced, emissions per ton-km). These modes of green finance standards may or may not be technology-agnostic. These types of standards require a high sophistication for the verification of green finance, as the financial institutions have to measure, report and validate the environmental impacts of their investments.

To test whether these qualitatively identified green finance models can be confirmed quantitatively, we identify the frequency of a collection of words that are descriptive for input, process, and output, which were informed based on Timmermans and Epstein [19] (see Table 3).

**Table 3.** Keywords for process, input and output standards.

| Process | Input | Output |
|---|---|---|
| EIA | catalogue | $CO_2e$/kwh |
| environmental impact assessment | list | emissions |
| safeguards | | threshold |
| project evaluation | | $CO_2$ |
| reporting | | outcome |
| appraisal | | screening criteria |
| approval | | criteria |
| procedure | | |
| screening process | | |

### 5.4.1. Hypothesis 4a: Actor–Model Relationship

We first test for association between model and actor using the Chi-square test. The result shows that there is a significant difference between actor and model ($p < 0.001$, Cramer's V = 0.235). Regulators are significantly positively associated with output standards and negatively associated with process standards. Those relationships contribute to 86.7 percent of the total Chi-square score and thus account for most of the difference between expected and observed values. We also find a statistically significant negative relation between output standards and development finance institutions and multilateral associations, and process standards and development finance institutions (see Figure 6). Given the influence of regulator residuals, we re-evaluate the statistical significance of the association without the effects of the regulator, given its significant contribution to the Chi-square score. Excluding regulators, the results remain significant ($p = 0.047$, Cramer's V = 0.054).

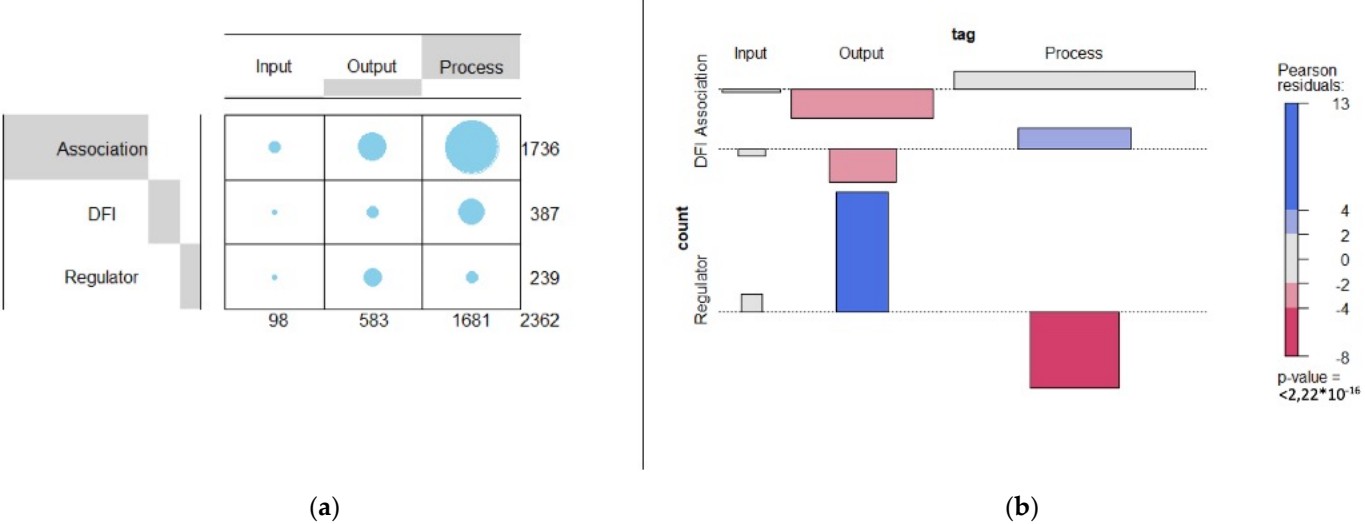

(**a**) (**b**)

**Figure 6.** Fisher's Exact Test for Hypothesis 5a—actor and model. It shows (**a**) the frequency of input-, output- and process-related terms issued by different issuers and (**b**) calculates the Pearson's residual, showing for example that "output"-related green finance standards are significantly positively correlated with "regulator", and significantly negatively correlated with developing finance institutions (DFIs).

### 5.4.2. Hypothesis 4b: Region–Model Relationship

We next test for association between focus and region using the Chi-square test. The result shows that there is a significant difference between region and focus ($p < 0.001$, Cramer's V = 0.059) (see Figure 7). There is a statistically significant positive association between China and input model, which contributes to 28 percent of the total Chi-square score. Alternatively, there is a statistically significant negative association between the EU and input model and a statistically significant positive association between the EU and

output model, which together contributes to 37.7 percent of the Chi-square score. Finally, emerging economies have a statistically significant negative association with output models, accounting for 19.8 percent of the Chi-square score.

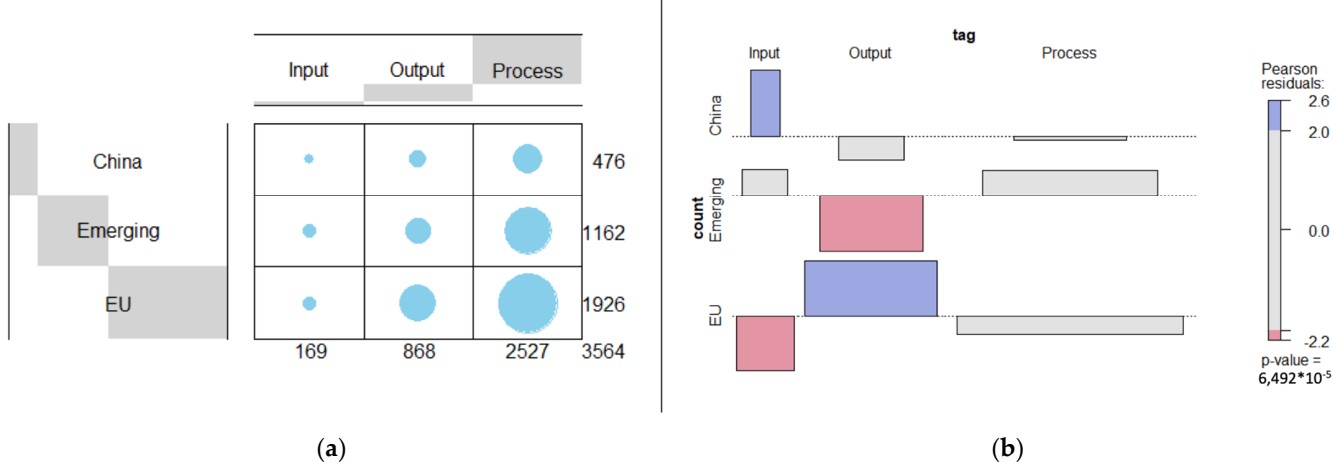

(**a**)                              (**b**)

**Figure 7.** Fisher's Exact Test for Hypothesis 4b—jurisdiction and model. It shows (**a**) the frequency of input-, output- and process-related terms issued in different jurisdictions (China, emerging economies, EU) and (**b**) calculates the Pearson's residual, showing for example that "input" models green finance standards are significantly positively correlated with China, and significantly negatively correlated with the EU.

### 5.4.3. Hypothesis 4c: Time–Model Relationship

Finally, we test for association over time using the Chi-square test. The result shows that there is a significant difference between time periods and model ($p < 0.001$, Cramer's V = 0.199). Process models were more prominent over the 2000–2010 period but have since become less prominent by 2015. Together, process residuals account for 24.2 percent of the total Chi-square score. Comparably, output models remained significantly less prominent between 2000–2010 and 2010–2015, before seeing a significant positive association post-2015. Output based residuals account for 71.1 percent of the total Chi-square score. Finally, between 2010 and 2015, input models became slightly more prominent, but that relationship did not carry over to the post-2015 period (see Figure 8).

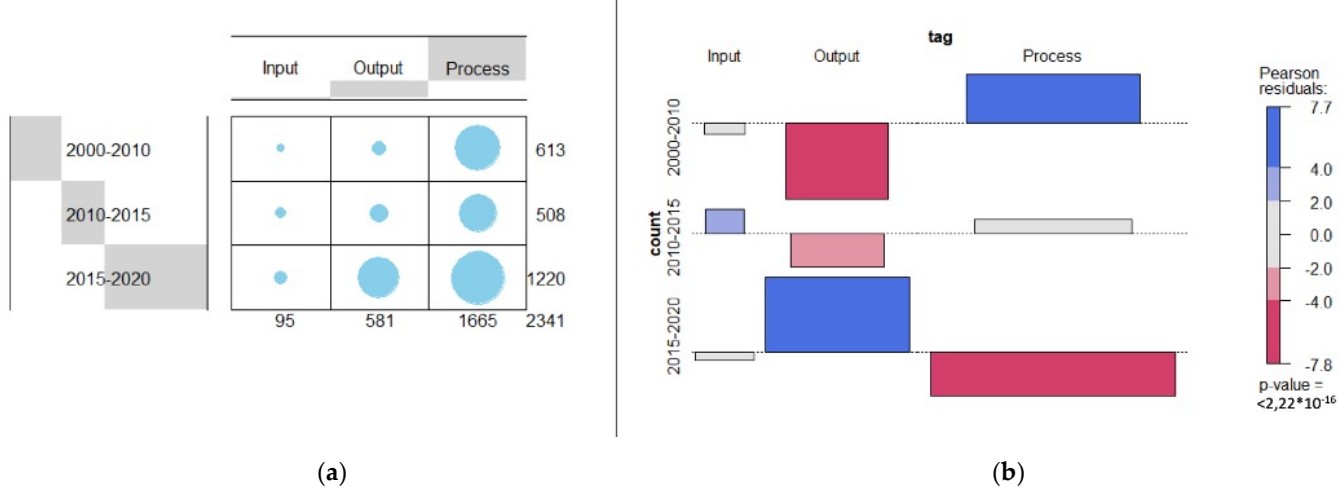

(**a**)                              (**b**)

**Figure 8.** Fisher's Exact Test for Hypothesis 4c—time and model. It shows (**a**) the frequency of input-, output- and process-related terms issued across time and (**b**) calculates the Pearson's residual, showing for example that "output" models of green finance standards have been significant since 2015, while from 2000 to 2010, "process" models were particularly significant.

As can be seen in Figure 9, many Chinese green finance standards would be considered input standards providing labels. In contrast, most sustainable finance standards we collected would be considered process standards, with only a few being output standards (e.g., EU Taxonomy).

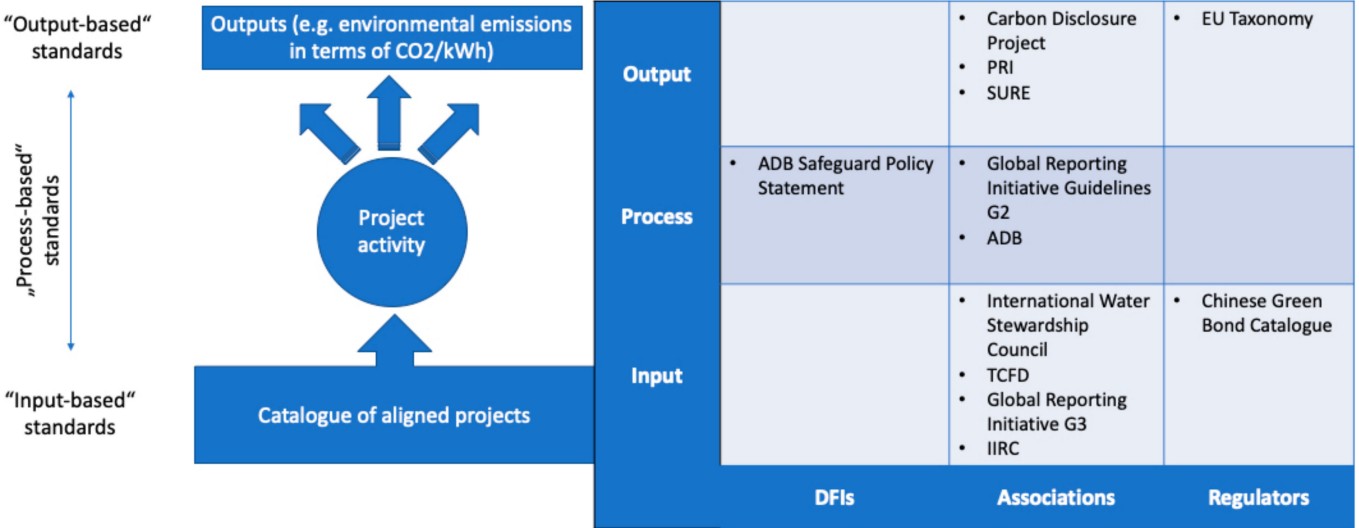

**Figure 9.** "Input-", "Process-", and "Output-" based sustainable finance standard models with examples (own depiction).

## 6. Discussion

Green finance standards serve to mobilize and apply finance for green development by addressing many different aspects regarding financial instruments (e.g., bonds, loans, equity), regarding finance lifecycle (raising capital, project finance, trading), and regarding green aspects (e.g., biodiversity, pollution, greenhouse gas emissions). With growing awareness of the responsibility of the financial sector to contribute to green development—the "financialization" of green development [3,4], a plethora of green finance standards have been issued and applied by governments, financial institutions, NGOs and associations over the past few years. However, little has been researched to understand why and how green finance standards develop in different ways leading to different outcomes, structures and applications of those standards. In other words, little is known about the nature of green finance standards.

In this paper, we applied institutional theory and standards theory to build four major hypotheses on the development of green finance standards to understand and analyze the nature and thus the evolution and application of global green finance standards. We test our four hypotheses empirically using text analysis and statistical methods of 86 green finance standards of three different types of economic governance systems (a market-economy in the EU, state-led economy in China and a weak institutional environment in emerging markets) issued between 1998 and 2020.

In our first hypothesis, we postulate that depending on the country's economic governance system, green finance standard development will be driven on different levels (e.g., government-led, market-led, "invisible-hand"-led) and accordingly applied differently depending on the governance system. We find that depending on the governance environment, applied standards are developed on different levels in the EU, China and emerging economies: in the government-led governance system China, government-led standards are widely issued while voluntary application of market-driven finance-standards is negligible; in the EU, voluntary standards are developed and widely applied, while government-led standards have been trailing in their development and exist alongside market-driven standards; in emerging economies, relevant green finance standards are issued by developing finance institutions, with governments and voluntary standards developed and applied by local institutions at a later stage.

In Hypothesis 2, we look at the ecological focus of green finance standards and postulate that the focus of the green finance standards will be different depending on the most-pressing local environmental problem. Using text analysis, we analyze the frequency and relevance of keywords related to "biodiversity", "pollution" and "climate" in the green finance standards. We find that development finance institutions place a larger focus on biodiversity than governments and market standards, the latter focusing more on climate issues. Pollution has not been found to be a particular focus area for any actor, which we found surprising (e.g., as China included "clean coal" into its green finance system to combat air pollution, but not to combat climate change). This might be due to the choice of keywords. We also looked at the development over time and region and find no significant difference between periods and focus.

In Hypothesis 3, we test whether green finance standard development will accelerate the broader the common understanding of environmental issues ("objectivation"). We tested using amongst others a Pearson's product–moment correlation test. We find relatively few standards issued before 2010 and that developments in green finance standards have been accelerating since 2016. We infer a correlation with the signing of the Paris Agreement in 2015 that brought a broad recognition of the finance and climate nexus and thus accelerated the financialization of green development.

In Hypothesis 4, we test whether different models of green finance standards are evolving in line with the models suggested by Timmermans and Epstein [19]. Based on quantitative and qualitative analysis of the standards, we can confirm that three models of green-finance standards have evolved: (1) input-driven models that focus on labelling of specific "green" technologies; (2) output-driven models that provide thresholds for environmental emissions, etc.; and (3) process-driven models that provide safeguards and performance standards to ensure necessary managerial steps are taken to take into account environmental risks. We find that regulators are significantly positively associated with output standards and negatively with process standards, with Chinese green finance standards being input-driven models, while the EU's standard is output-driven. Emerging markets have a significant negative association with output-driven models. When testing over time, we find that between 2000 and 2010, process models were more prominent, between 2010 and 2015 input models become more prominent, and since 2015 output models have become more prominent.

The research has allowed us to highlight how in some of the most important markets for green finance different green finance standards are both emerging and necessary— after all "we coexist in a world filled with standards, but not a standard world" [19]. These differences depend on several factors: First, the type of agents with power, where in market economies non-government actors will play a bigger role, while in government-centric economies like China, the government as an agent must play the most important role. Second, the maturity of the market, including the institutional capacity, are relevant for the development and application of green finance standards. With different actors and different jurisdictions, we show that the focus area (i.e., pollution control, GHG emissions, biodiversity) of green finance standards vary depending on the local context. As expected, to accommodate the differences, we could prove that different models of green standards have emerged: output-based standards, process standards, and input-based standards. This research therefore also highlights that a "one-size-fits-all" green finance standard is currently difficult—which also precludes the answer to the question "which green finance standard is the best", as the "best" is locally specific. However, the results provide the basis to research possibilities to combine different models of green finance standards, e.g., input, process and output based standards, each with their individual strengths and application: input standards and labels have relatively low transaction costs due to their simple lists of aligned projects; process-based standards provide flexibility to adjust investments and financing better to different circumstances as they are less rigid; output-based standards with their de-facto measurement of environmental impacts to avoid greenwashing [22]. A combination of the standards adjusted to the local circumstances, after all, could help

to accelerate de-facto application of green finance with a de-facto shift of financing and investment into a green economy that protects climate, biodiversity and is pollution-free.

**Author Contributions:** Conceptualization: C.N., T.D., O.W.; literature review: C.N.; hypothesis development: C.N., T.D., O.W.; data collection and data sorting: C.N., T.D.; statistical hypothesis testing: T.D.; hypothesis evaluation: T.D., C.N., O.W.; discussion: C.N., T.D., O.W. All authors have read and agreed to the published version of the manuscript.

**Funding:** This research received no external funding.

**Institutional Review Board Statement:** Ethical review and approval were waived for this study as this study focuses on technical standards for finance and did not involve animals.

**Informed Consent Statement:** Not applicable.

**Data Availability Statement:** Data supporting reported results can be found at www.green-bri.org/green-finance-codes (accessed on 13 May 2020).

**Acknowledgments:** We are grateful for the feedback received at the presentation of initial findings at the 2020 GRASFI Conference hosted by Columbia University. We are further grateful for the support by various colleagues in reviewing initial versions of the paper and giving valuable feedback, particularly Yao Wang, Mathias Lund Larsen and Ben Caldecott.

**Conflicts of Interest:** The authors declare no conflict of interest.

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
