# Peer review of "The Nature of Global Green Finance Standards—Evolution, Differences, and Three Models"

_sustainability, doi:10.3390/su13073723_

Round 1
Reviewer 1 Report
This paper applies institutional theory to analyze the development of green standards in three contexts: a liberal context, a state-controlled context and an emerging market context. It presents interesting data on the multiplication of green finance labels. While I enjoyed reading the paper there are a few issues which would definitely need to be tackled in a revised version:
- I think the paper should better describe the architecture of Financial systems in the three studied contexts (in particular the Chiniese vs. European context): the role of banks, markets, the role of the Central Bank, green finance regulations etc....
- The paper should give information about the ecological transition strategy in each zone prior to studying the role of labels and how they are constructed
- The papers currently analyzes label construction based on keywords, leaving aside the key dimension from a Sustainability standpoint: do labels have requirement based on scientific data (e.g. CO2emissions) or are there mere "commitments" from market players to "do better"?
- I think the authors should link their discussion with te "financialization of green finance" and greenwashing literature: are labels a tool to operate a real economic transformation, or mere "brands"/symbols? (see Chiapello, 2020, Nyström et.al, 2019 and grey literature such as https://wrm.org.uy/browse-by-subject/mercantilization-of-nature/financialization-of-nature/)
- How were the key words selected in the analysis?
- The authors should have a look at the sociological and management literature to better decribe the label construction process (see for instance Giamporcaro et.al, 2020 in the European)
- There are several typos and irrelevant paragraphs in the text (for instance the last paragraph of the introduction); these would have to be checked and corrected
References
Chiapello, Ève, 2020. "Stalemate for the financialization of climate policy," economic sociology_the european electronic newsletter, Max Planck Institute for the Study of Societies, vol. 22(1), pages 20-29.
S Giamporcaro, JP Gond, N O'Sullivan - Business Ethics Quarterly, 2020. Orchestrating governmental corporate social responsibility interventions through financial markets: The case of French socially responsible investment
Nyström M, Jouffray JB, Norström AV, Crona B, Søgaard Jørgensen P, Carpenter SR, Bodin Ö, Galaz V, Folke C. Anatomy and resilience of the global production ecosystem. Nature. 2019 Nov;575(7781):98-108. doi: 10.1038/s41586-019-1712-3. Epub 2019 Nov 6. PMID: 31695208.
Author Response
Comment 1: “I think the paper should better describe the architecture of Financial systems in the three studied contexts (in particular the Chinese vs. European context): the role of banks, markets, the role of the Central Bank, green finance regulations etc....”
Response: Thank you for this idea. We agree with this comment and added this distinction and description accordingly:
- In chapter 3: We added several paragraphs and included references to better describe market-versus bank-based systems (e.g. Demirguc-Kunt and Levine 1999, Tadesse 2002).
- We described the structure and functions of different institutions of the financial markets in the different systems in China, the EU and also in emerging markets (e.g. the role of the ECB, PBOC), as well as adding some details on the sizes of the markets, particularly in chapter 3 on the data, as well as in chapter 4 about the results.
- We also highlighted that all three economic governance models (EU, China, developing countries) tend to be bank-based systems.
Comment 2: “The paper should give information about the ecological transition strategy in each zone prior to studying the role of labels and how they are constructed”
Response: We agree with this comment and added more information about the ecological transition strategy:
- In chapter 3 (data), we added several references to describe historical and current developments of the ecological transition strategy, particularly for China and the EU based on official government documents and previous research.
- For example, we describe how the EU included climate change in its sustainable development strategy in 2001 and biodiversity in 2007, while it called upon business, NGOS and citizens to become involved in the sustainable development transition. We also improved the information on the development of the EU Taxonomy..
- China, meanwhile, focuses on its “three battles” of air pollution, financial stability and poverty alleviation, and is aiming to “win the battle for the blue skies”. Thus, China’s environmental policies have focused on air pollution and accordingly green finance labeled “clean coal” to reduce air pollution as green finance in its green bond and green industry catalogue. For climate, China issued a climate finance guidance in November 2020.
Comment 3: “The papers currently analyzes label construction based on keywords, leaving aside the key dimension from a Sustainability standpoint: do labels have requirement based on scientific data (e.g. CO2emissions) or are there mere "commitments" from market players to "do better"?”
Response: The reviewer raises a crucial point in green finance (also in comment 4): what is the impact of green finance on green development, and in consequence: how much of green finance standards are requirements versus mere “relabelling” of previous commitments - which might ultimately lead to greenwashing.
Much research has already been devoted to the greenwashing, also in this journal (e.g. Pimonenko et al, 2020).
We added some specifications to look at the literature on labelling and greenwashing. We highlight, for example, that the green-washing literature (e.g. Berensmann and Lindenberg (2016), Harlan (2020) finds that common green finance standards would allow for less greenwashing.
Common standards, as we describe in this paper, require an understanding of the nature of green finance standards, which is the core of our study: we do not try to answer which green finance standard is “best” for green development, but study how green finance standards are local, context and agent specific – each leading to green finance standards with their specific requirements and ambitions on what green development is.
Therefore, we find that a specific financing activity might be greenwashing in one jurisdiction, it is not in the other jurisdiction. To give an example: investing in clean coal is greenwashing in the eyes of climate protection, while it is not necessarily in the eyes of Chinese green finance academics who focus on pollution reduction.
To specify this issue and draw attention to the greenwashing and labelling (without additionality) issue, we added specifications and references in the introduction, hypothesis 4 and the conclusion.
Our paper also aims to provide a clear distinction between input and output standards, where input standards provide one form of labels (albeit, also output and process standards can provide labels) versus output standards take the de-facto green development contribution into consideration.
Comment 5: “I think the authors should link their discussion with the "financialization of green finance" and greenwashing literature: are labels a tool to operate a real economic transformation, or mere "brands"/symbols? (see Chiapello, 2020, Nyström et.al, 2019 and grey literature such as https://wrm.org.uy/browse-by-subject/mercantilization-of-nature/financialization-of-nature/)
Response: The reviewer’s question is highly relevant: what is actually the efficacy of green finance standards. The study focuses on the research on the evolution, types and application of green finance standards, and less on the environmental impact and thus the greenwashing part of the standards. We hope, that our study can provide a basis for further research on the efficacy of green finance standards (see response to comment 4).
To further specify this issue, we included the concepts of greenwashing and labels Chiapello, 2020 and Nyström, 2019 (e.g. also in the description of the “input” standard).
We also included the concepts of financialization of green development (nature, climate) throughout the paper, e.g. in hypotheses 3 on the proliferation of green finance standards..
Comment 6: “How were the key words selected in the analysis?”
Response: This is an important question and we added details on the selection of keywords for the text analysis:
- We specified that for the “green” elements of biodiversity, climate and pollution we analyzed the keywords from CBD (biodiversity), UNFCCC and IPCC (climate) and WHO (pollution)
- We added details on the selection of keywords for the analaysis of process, input and output based standards based on the literature (e.g. Timmermans and Epstein, 2010).
Comment 7: “The authors should have a look at the sociological and management literature to better describe the label construction process (see for instance Giamporcaro et.al, 2020 in the European)”
Response: Thank you for the literature. We included more information on the label construction process, particularly in the description of the input standards hypotheses.
Comment 8: There are several typos and irrelevant paragraphs in the text (for instance the last paragraph of the introduction); these would have to be checked and corrected
Response: Thank you for the hint. We checked the document again for typos and deleted e.g. the last paragraph of the introduction
Reviewer 2 Report
Enlarge references to literature on EU sustainable finance action plan.
Revise various misprints.
Author Response
Thank you for your support in reviewing and improving our paper!
Comment 1: "Enlarge references to literature on EU sustainable finance action plan."
Response: We expanded the literature and information provided on the EU sustainable finance action plan and the overall development of both China's and the EU's green development agenda in the data section (chapter 3).
Round 2
Reviewer 1 Report
The authors have responded all comments very diligently.